# Widespread variation in heat tolerance and symbiont load are associated with growth tradeoffs in the coral *Acropora hyacinthus* in Palau

**Brendan Cornwell[1]\*, Katrina Armstrong[1], Nia S Walker[1], Marilla Lippert[1], Victor Nestor[2], Yimnang Golbuu[2], Stephen R Palumbi[1]**

[1]Department of Biology, Hopkins Marine Station of Stanford University, Pacific Grove, United States; [2]Palau International Coral Reef Center, Koror, Palau

**ABSTRACT** Climate change is dramatically changing ecosystem composition and productivity, leading scientists to consider the best approaches to map natural resistance and foster ecosystem resilience in the face of these changes. Here, we present results from a large-scale experimental assessment of coral bleaching resistance, a critical trait for coral population persistence as oceans warm, in 221 colonies of the coral *Acropora hyacinthus* across 37 reefs in Palau. We find that bleaching-resistant individuals inhabit most reefs but are found more often in warmer microhabitats. Our survey also found wide variation in symbiont concentration among colonies, and that colonies with lower symbiont load tended to be more bleaching-resistant. By contrast, our data show that low symbiont load comes at the cost of lower growth rate, a tradeoff that may operate widely among corals across environments. Corals with high bleaching resistance have been suggested as a source for habitat restoration or selective breeding in order to increase coral reef resilience to climate change. Our maps show where these resistant corals can be found, but the existence of tradeoffs with heat resistance may suggest caution in unilateral use of this one trait in restoration.

**\*For correspondence:**
bcornwel@stanford.edu

**Competing interest:** The authors declare that no competing interests exist.

## Introduction

Climate change is increasingly shifting species ranges, altering ecosystem dynamics, and generating strong selection differentials in wild populations (*Chen et al., 2011*; *Logan et al., 2014*; *MacLean and Beissinger, 2017*; *Wiens, 2016*; *Parmesan and Yohe, 2003*; *Chen et al., 2011*; *Lenoir and Svenning, 2015*). Against this backdrop, there is an accelerated focus on characterizing the adaptive mechanisms that increase resilience to climate stressors in natural communities (*King et al., 2018*; *Walsworth et al., 2019*; *National Academies of Sciences Engineering and Medicine, 2019a*).

The practical importance of identifying populations that are locally adapted to climate stress is rooted in the possibility that populations already harboring stress-tolerant individuals might be used in restoration projects or in assisted evolution efforts to enhance the resilience of vulnerable populations (*Aitken and Whitlock, 2013*; *Mascia and Mills, 2018*; *Seddon et al., 2014*; *National Academies of Sciences Engineering and Medicine, 2019b*). *Walsworth et al., 2019*, and *McManus et al., 2021* used eco-evolutionary models to show that evolutionary responses to climate change stress could lead to higher levels of stable persistence than ecological models without evolution. Identifying source populations with such phenotypic variance could become especially critical for foundation species such as corals, forest trees, grasslands, and seagrasses (*Franks et al., 2014*; *Hodgins and Moore, 2016*; *Morikawa and Palumbi, 2019*). Characterizing the distribution of stress-tolerant

individuals could also be the basis for investigating the mechanisms leading to heat stress resistance and, ultimately, predicting how these populations will respond to climate change.

In addition to prioritizing conservation efforts, identifying geographic locations harboring stress-tolerant individuals allows researchers to begin assessing how plasticity and local adaptation shape individual phenotypes. Strategies for promoting populations that are resilient to climate change will depend on the relative strength of these forces, which will aid in choosing individuals for transplantation, seed sources, or selective breeding programs (see *van Oppen et al., 2015*).

However, an important caveat is the possibility that stress resistance carries an associated cost. For example, studies of grass populations show that drought-resistant individuals grow more slowly (*Blumenthal et al., 2021*). Theory suggests that locally beneficial alleles (e.g., those that confer stress tolerance) can remain polymorphic in a population if they are selectively deleterious in other environments (*Levene, 1953*). Thus, characterizing the costs and benefits of resisting climate change is fundamental for predicting if stress resistance will increase in frequency across generations, and how increased resistance will impact the diversity and productivity of an ecosystem (*Blumenthal et al., 2021*).

Coral populations can be differentially adapted to high temperatures across extensive geographic ranges (*Dixon et al., 2015*; *Berkelmans and Willis, 1999*; *Sawall et al., 2015*) or across different local microclimates (*Palumbi et al., 2014*; *Morikawa and Palumbi, 2019*). However, tradeoffs in heat tolerance have been more difficult to identify (*Bay and Palumbi, 2017*; *Shore-Maggio et al., 2018*; *Muller et al., 2018*) and remain a key concern of researchers and managers (*National Academies of Sciences Engineering and Medicine, 2019a*; *National Academies of Sciences Engineering and Medicine, 2019b*).

Here, we provide the first archipelago-wide view of regional and small-scale variation in heat tolerance in corals by conducting standardized heat stress tests on the tabletop coral *Acropora hyacinthus* across 37 reefs in Palau. By mapping and testing hundred s of individual corals across reefs with different microhabitats, we generated a fine-scale heat tolerance map and compared it to temperature, depths, and annual growth in the field. The results show a wide distribution of heat-tolerant colonies that are concentrated in – but not exclusive to – warmer patch reefs. However, we also find lower symbiont load in most heat-resistant corals, and that low symbiont levels are correlated with lower growth rates. The large inventory of heat-tolerant colonies across the archipelago could provide ample, local stocks of corals for natural evolution of heat tolerance. Yet, the possibility of a fundamental tradeoff between growth and bleaching resistance highlights the need to carefully consider the benefits and risks of intervention strategies focusing on a single trait.

## Results and discussion
### Geography of heat resistance

Across 221 colonies of the tabletop coral *A. hyacinthus* from 37 reefs in Palau, we found wide variation in bleaching susceptibility. In a simple 2-day standardized heat stress experiment, colonies ranged from retaining virtually all of their original symbiont load at 34–35°C (ca. 4–5°C above ambient temperatures) to less than 10 % at these temperatures (*Figure 1*). Reef regions with the most heat-resistant colonies have higher exposure to temperature extremes (>32 °C, *Figure 2C*), and the same pattern occurs among individual reefs (*Figure 2D*, Spearman's rank correlation for 32 °C, S = 3725, p = 0.0304; linear model $R^2$ = 0.1216, p = 0.02648; see supplemental appendix for data and analysis on reef temperature and *Figure 2—figure supplement 4* for this relationship using temperature thresholds of 31–35°C). Previous studies showed that bleaching-resistant individuals can inhabit a subset of microclimates such as shallow back reefs (*Oliver and Palumbi, 2011*) or the intertidal zone where large temperature swings are common (*Schoepf et al., 2015*). Our data extend this to warm lagoon patch reefs, a very common feature of complex reef ecosystems around the world, and even to some fore reef locations with high heat exposure, helping identify other possible targets for heat resistance prospecting.

Yet, we also find wide variation in heat resistance on individual reefs with cooler temperatures. For example, two patch reefs and one fore reef in the Northern Lagoon experience few warm water events but have high numbers of heat-resistant colonies (reefs 41, 42, 65, *Figure 2D*), Across all reefs, bleaching-resistant colonies are widespread: 24 of 37 reefs harbor at least one colony that falls in the

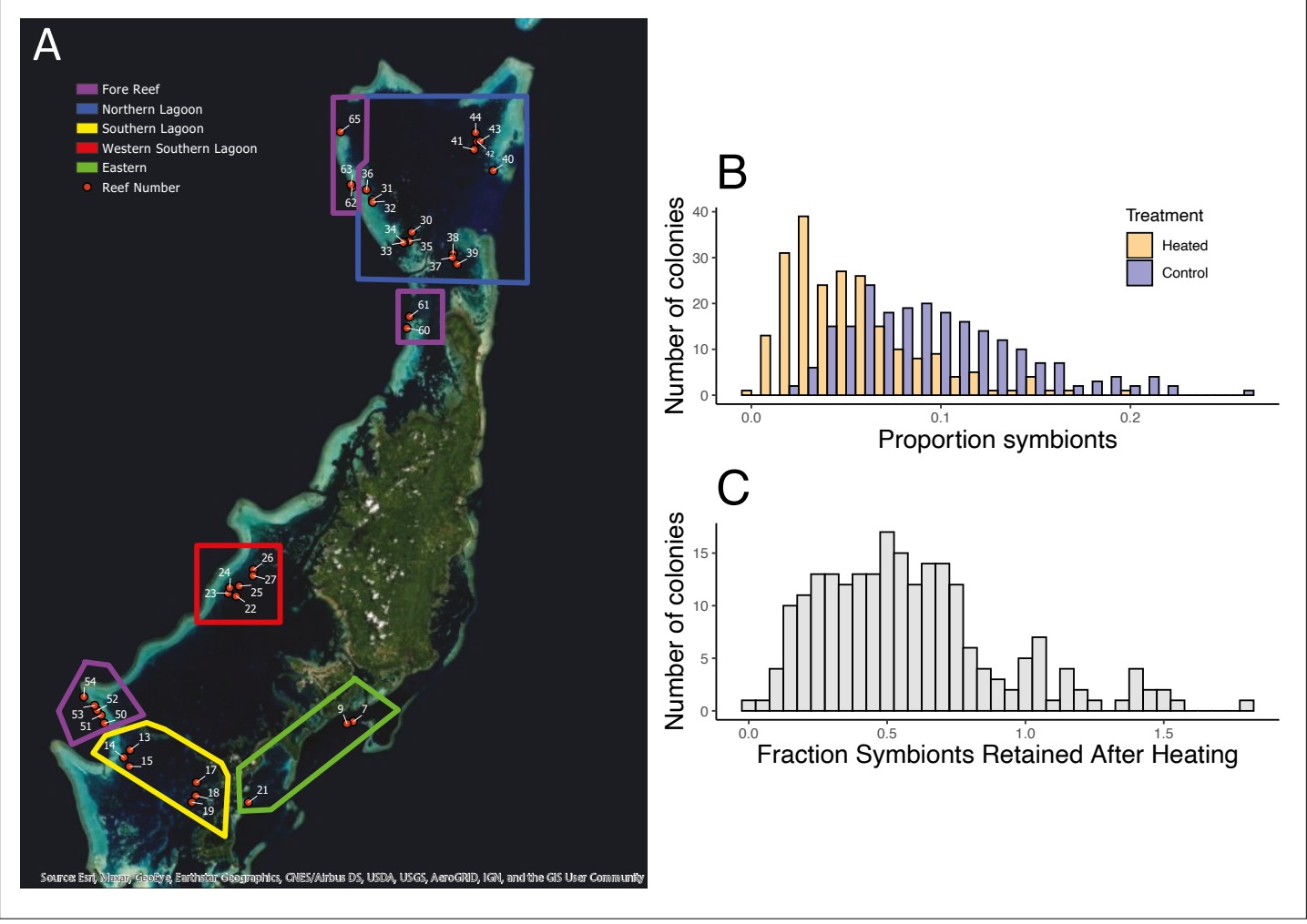

**Figure 1.** Geographic distribution of reefs and bleaching responses after experimental warming. (**A**) Map of 39 reef locations surveyed, arranged in groups in the North (blue), West (red), East (green), and South (yellow). Ten reefs that are outlined in purple are at fore reef locations. (**B**) Mean proportion of symbionts in tissues from corals before and after heating. (**C**) The fraction of symbionts retained after heating across all 221 colonies. Accompanying source data are available as *Figure 1—source data 1*data.

The online version of this article includes the following figure supplement(s) for figure 1:

**Source data 1.** Colony by colony bleaching response to experimental warming.

**Figure supplement 1.** Range of symbiont cell concentration by visual bleaching score category.

**Figure supplement 1—source data 1.** Symbiont density as measured by flow cytometry and visual bleaching score.

**Figure supplement 2.** Colonies with high symbiont retention after heat stress tend to have lower levels of symbionts (gray bars), compared to symbiont load across all colonies (blue bars).

**Figure supplement 2—source data 1.** Colony by colony symbiont load pre- and post-experimental bleaching.

top quartile for bleaching resistance (*Figure 2A and B*; *Video 1*). Thus, well-defined conservation strategies that focus on wide regions that have historically experienced higher temperatures such as the Red Sea (*Krueger et al., 2017*) could overlook smaller, local geographic areas where bleaching resistance is more common. Even fore reefs, which include fewer high-temperature microclimates than back reefs or patch reefs, harbor enough heat resistance in Palau to significantly increase the inventory of such corals across the archipelago.

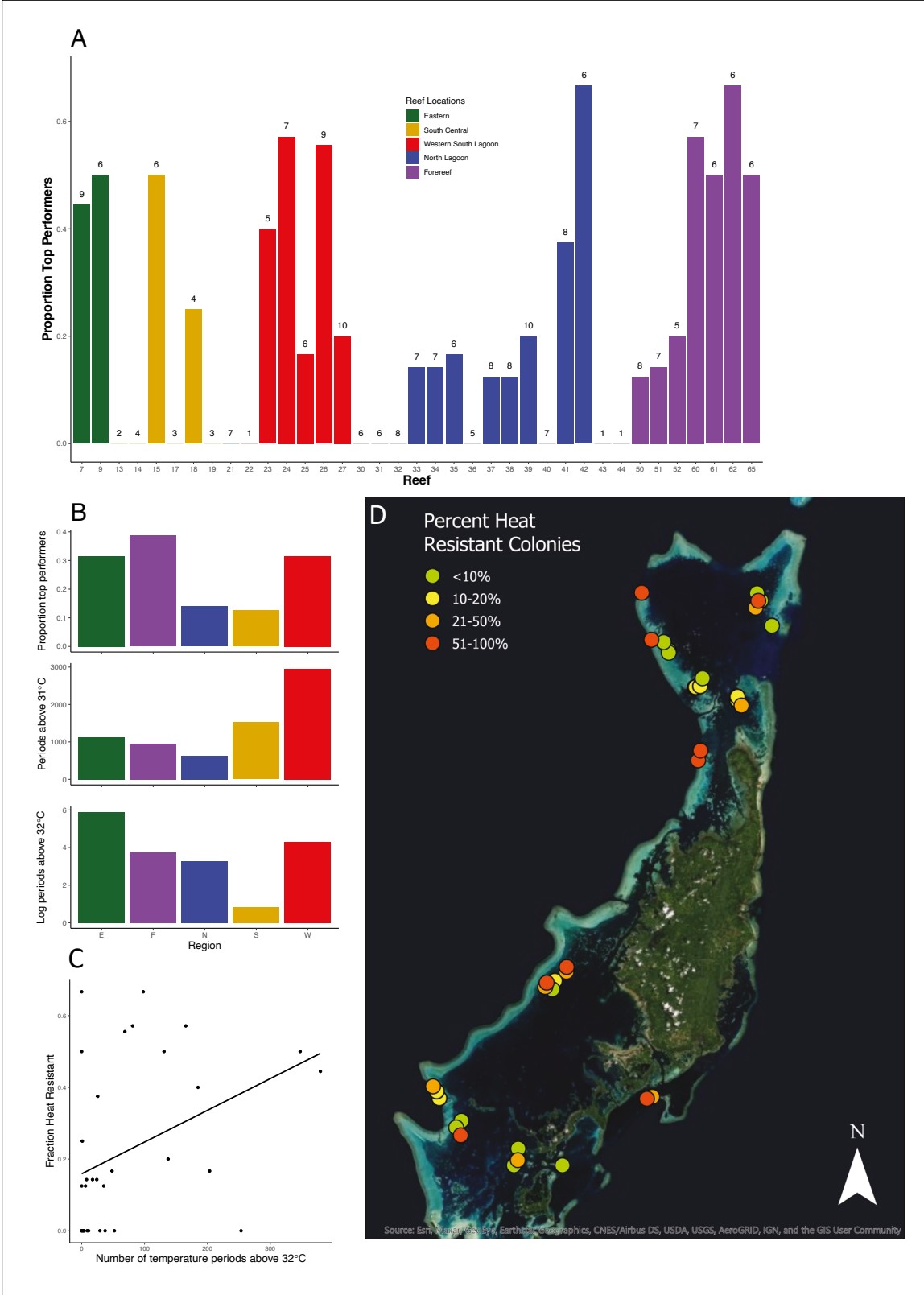

**Figure 2.** Location and prevalence of heat-resistant colonies by reef and region. (**A**) Location of corals in the top 25 % of values for symbiont retention. See *Figure 1A* for reef locations. Numbers above each reef label are the number of colonies sampled from that reef. (**B**) Location of corals that are in the top 25 % in symbiont retention by region and associated mean temperatures (note log scale for 32°C). (**C**) Plot showing the relationship between the average number of 10 min intervals above 32 °C on a reef and the fraction of colonies on that reef in the top 25 % of values for symbiont retention.

*Figure 2 continued on next page*

*Figure 2 continued*

(**D**) The distribution and frequency of bleaching-resistant colonies across the Palauan archipelago. Colors correspond to the frequency of highly heat-resistant corals found in this survey on each reef. Accompanying source data are available as *Figure 2—source data 1*.data.

The online version of this article includes the following figure supplement(s) for figure 2:

**Source data 1.** Reef locations, number of extreme temperature events and proportion of bleaching resistant individuals.

**Figure supplement 1.** The percentage of days with temperatures above 30 °C, 30.5 °C, and 31 °C for four regions encompassed by the vertices of the yellow lines.

**Figure supplement 1—source data 1.** Regional temperature profiles using remote sensing data.

**Figure supplement 2.** Mean symbiont load of *Acropora hyacinthus* colonies across Palau.

**Figure supplement 3.** Average colony symbiont load (number of symbiont cells per coral cell) averaged across colonies from different reefs.

**Figure supplement 3—source data 1.** Mean symbiont load for colonies inhabiting each reef.

**Figure supplement 4.** Proportion of heat-resistant colonies on each reef as a function of the number of extreme temperature events, depicting the relationship when the threshold is 31°C, 32°C, 33°C, 34°C, and 35°C (A–E, respectively).

**Figure supplement 4—source data 1.** Mean number of extreme temperature events recorded at each reef (thressholds 31-35°C).

## Bleaching intensity, symbiont load, and growth

Symbiont load in individual colonies was bimodal in our non-heated control nubbins and had a considerable range (*Figure 1B*): the higher group of corals showed 11–20% symbiont cells per counted coral cell, whereas the lower group was centered on symbiont levels of 5–6% (*Figure 1B*; dip test, n = 221, p = 0.0167). Variation in load was high within reefs as well as between reefs (average standard deviation within reefs = 0.041, compared to 0.045 for the whole data set), reflecting marked variation among colonies close to one another. Some reefs had significantly higher loads (reefs 27, 51, 65; ANOVA, p = 0.005) but these were not related to temperature, depth, latitude, or other environmental correlates ($R^2$ <0.005).

However, variation in symbiont load was inversely correlated with symbiont retention after our standard heat test (*Figure 3A*, p = $5.55 \times 10^{-6}$, $R^2$ = 0.08597). Overall, the most bleaching-resistant colonies began with lower symbiont levels than the most bleaching sensitive colonies (8.0% versus 11.3%, *Table 1*).

We returned to the marked colonies in July 2019 to re-measure colony size after 1 year of growth in the field. Average colony linear extension (4.8 cm year$^{-1}$, standard deviation = 3.2 cm) and percent growth (5.96%, standard deviation = 4.1%) were similar to averages seen for these species in other locations (*Gold and Palumbi, 2018*). After removing size decreases from damage or disease (conservatively, decreases exceeding 10%), colonies with higher symbiont load have significantly higher growth rates (*Figure 3B*, $R^2$ = 0.026, p = 0.0398). We saw a similarly significant increase in growth in colonies with the highest symbiont load in a linear model with size, retention, and symbiont load as fixed effects ($R^2$ = 0.038, p = 0.0477). Comparing corals with higher versus lower than median symbiont load, growth was approximately twice as high, 5.6% and 2.7%, respectively.

## A tripartite coral phenotype

These data point to a complex interrelationship between three coral phenotypes: bleaching resistance, symbiont load, and growth potential. *Cunning and Baker, 2012*; *Cunning and Baker, 2014*, first showed that corals with high symbiont load were more susceptible to bleaching, and suggested that higher levels of molecules arising from damage to the symbiont photosystem – for example, 'reactive oxygen species' – in tissues

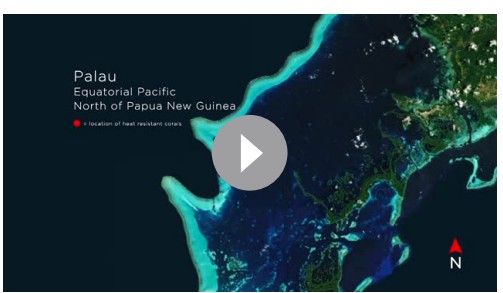

**Video 1.** Distribution of heat-resistant colonies in Palau. Animation depicting the approximate locations of bleaching-resistant colonies sampled for this study across the Palauan archipelago.
https://elifesciences.org/articles/64790/figures#video1

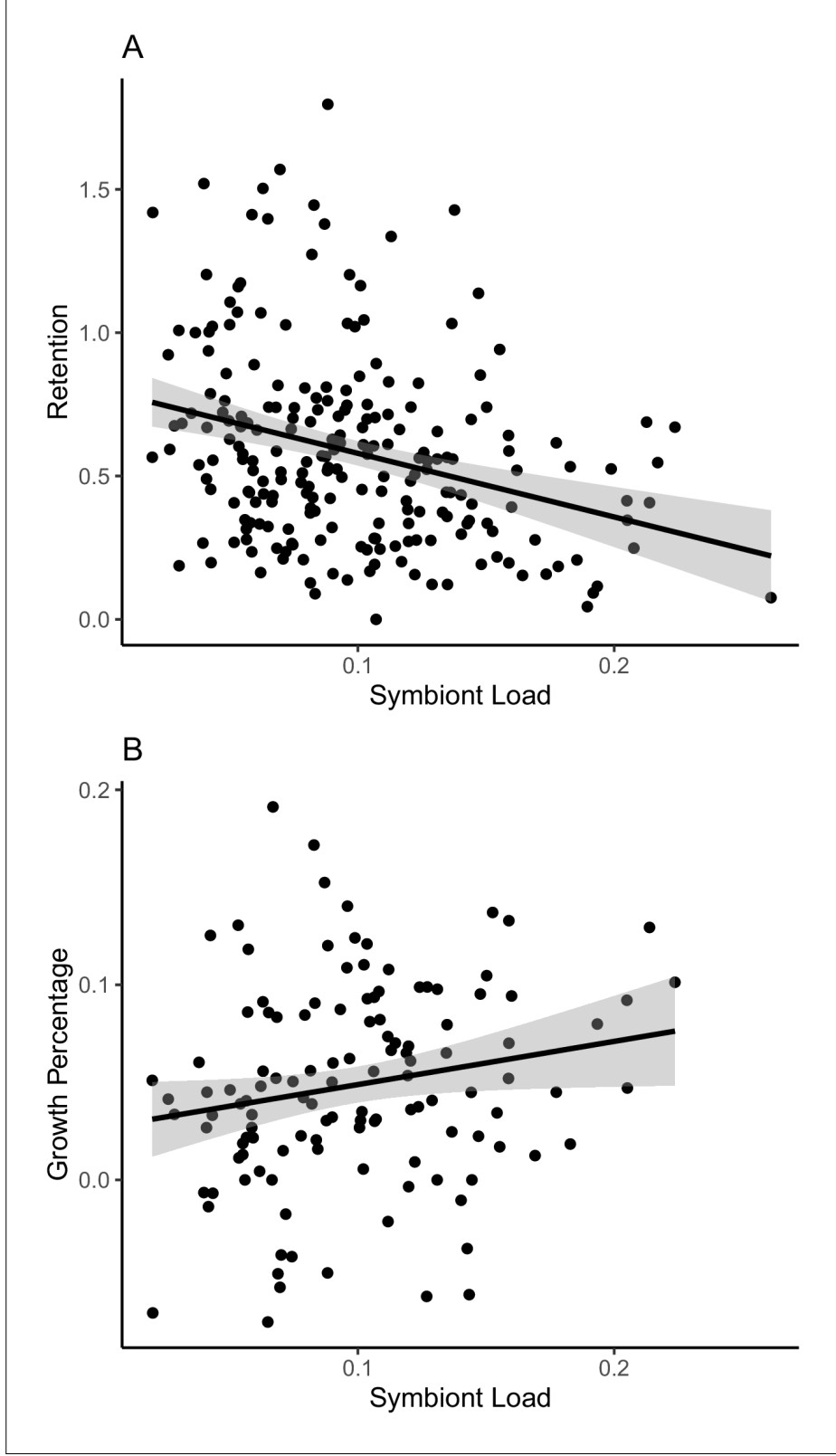

**Figure 3.** Relationships between symbiont retention and colony growth to initial symbiont load. (**A**) Mean starting symbiont density of *A. hyacinthus* colonies across Palau is negatively correlated with the fraction of symbionts retained after heat stress. Colonies with lower symbiont population densities (fraction of symbiont cells per coral cells) tend to show higher retention after 2 days of standardized heat stress ($r^2 = 0.080$, $p = 9.063 \times 10^{-4}$). (**B**) Annual

*Figure 3 continued*

growth (2018–2019) is higher for colonies with higher symbiont loads ($r^2$ = 0.026, p = 0.0398). Accompanying source data are available as *Figure 3—source data 1*.data.

The online version of this article includes the following figure supplement(s) for figure 3:

**Source data 1.** Colony by colony measurements of symbiont load, retention and growth.

---

with denser symbiont populations might explain this pattern. Other potential mechanisms, such as increased metabolic demands by the symbiont in warmer conditions (*Wooldridge, 2009*), could also favor hosts maintaining symbiont populations at low levels.

Likewise, bleaching resistance has been associated with slower growth, primarily between species. This has been seen in fast growing, heat-sensitive *Acropora* spp. and slow growing, heat-resistant *Porites* spp. (*Carpenter et al., 2008*). Fewer comparisons within coral species have been made. However, different genotypes of farm-raised Caribbean staghorn corals *Acropora cervicornis* varied negatively in growth and bleaching resistance, (*Ladd et al., 2017*).

The relationship between symbiont load and growth is more complex. *Wright et al., 2019*, showed a positive correlation between symbiont density and coral growth in lab experiments. But in other experiments, excess symbiont densities can sequester nutrients and result in less energy translocation to the coral (*Baker et al., 2018*). In particular, nitrogen limitation seems to affect symbiont photosynthesis, cell division, symbiont load, and heat resistance (*Morris et al., 2019*), with larger nitrogen to phosphorus ratios increasing the likelihood of bleaching.

Our study examines all three of these crucial facets of coral biology simultaneously, and for the first time shows how the role of symbiont density in both growth and bleaching might result in a negative tradeoff between them (*Figure 3A and B*). In this view, maintaining low loads of *Cladocopium* spp. symbionts could be a bet-hedging strategy where a coral grows at a slower rate but minimizes its risk of bleaching, in some ways analogous to the well-known tradeoff in growth versus heat resistance between *Cladocopium* spp. and *Durusdinium* spp. symbionts (*Lesser et al., 2013*). If this kind of tradeoff is widespread across corals, it may need to be taken further into account when heat tolerance is used as a criterion in reef restoration.

## Resilience tests in conservation and restoration

As climate change continues to reshape the seascape, conservationists and managers will need to quickly assess the vulnerability of populations to current and future temperatures, and design management plans that engineer resilience into populations under threat (*National Academies of Sciences Engineering and Medicine, 2019a*; *National Academies of Sciences Engineering and Medicine, 2019b*). Our study outlines a protocol using simple, standard heat stress tests to identify bleaching-resistant corals, which can help inform conservation strategies, in addition to informing future research that will be needed to effectively engineer climate resilience into future populations.

We found that across the Palauan archipelago, bleaching-resistant colonies inhabit a wide range of reef habitats and thermal environments. Future work should focus on determining how heritable genetic variation is in the host and symbiont, as well as plastic effects such as acclimation or nutrient response, shape this trait. In particular, if the heat resistance is heritable, one use of these data

**Table 1.** Comparison of bleaching-resistant and beaching-prone individuals.
Average results of bleaching experiments for corals that are in the top 25 % (highest quartile) of heat resistance rankings versus those in the bottom 25 % (lowest quartile).

| Rank | Control Symbiont proportion | Heated Symbiont proportion | Heated Retention | Avg depth | Temperature No. intervals above 31 °C | No. intervals above 32 °C |
|---|---|---|---|---|---|---|
| Top 25% | 0.080 | 0.082 | 1.041 | 0.954 | 1703 | 114 |
| Bottom 25% | 0.113 | 0.023 | 0.22 | 1.062 | 1714 | 79 |

The online version of this article includes the following source data for table 1:

**Source data 1.** Symbiont load and retention for all colonies.

could be in protection of reefs that already have large populations of bleaching-resistant corals. For example, patch reefs on the western edge of Palau's southern lagoon sit 1–10 km behind the barrier reef and form a micro-archipelago of shallow water habitats each no more than a few 100 m across that heat up at low tide. Protecting this area from further harm due to overfishing, habitat destruction or sedimentation could be an efficient way to preserve a large number of heat-resistant colonies for use in future interventions, or as a seed source after future bleaching events.

A second use is in assisted migration – transplanting heat-resistant corals to other habitats so that they can inject heat resistance genes into local populations. *Bay and Palumbi, 2017*, modeled adding 1–5% heat-resistant corals to a cool-adapted population in the Cook Islands and found that this could help prevent population extinction in some future $CO_2$ emissions scenarios. However, this model of selection and others (e.g., *Walsworth et al., 2019*; *McManus et al., 2021*) do not take into account the growth tradeoff we see here.

## Conclusions

There is an increasing call to renew ecosystems with future-adapted populations rather than restore them with populations adapted to previous conditions (e.g., *O'brien et al., 2007*). These management plans are advanced by standardized stress testing, rapid data collection, and extensive geographic surveys. By generating the first archipelago-wide map of coral heat resistance, we have shown a surprisingly wide distribution of heat-resistant colonies in some unexpected reef regions. Potential tradeoffs between bleaching resistance and other important phenotypes suggest caution in strategies of reef protection and assisted evolution that help heat tolerance at the cost of other key features like growth. Small-scale environmental variation (on the order of 1–10's of km) leading to exceptionally warm microclimates may have generated phenotypic variation in stress tolerance among many coral species, which could become an important asset in managing these reefs in the future.

## Materials and methods

We tagged 10 colonies of *A. hyacinthus* on each of 40 reefs across the Palauan archipelago in October 2017, on half of these individuals we also secured a HOBO data logger (Onset, MA) that recorded the temperature every 10 min. In the summer of 2018, data loggers were recovered, the largest diameter of each colony was measured, and a branch of the colony was sampled and transported back to the Palau International Coral Reef Center (PICRC) wrapped in seawater doused bubble wrap stored in a cooler. Upon return, corals were placed in a running seawater tank where they recovered overnight. In the morning, nubbins for each colony were divided into five pieces, two became control nubbins which remained at 30 °C for the duration of the experiment and three were subjected to experimental heating. The experimental heat treatments ramped from 30 °C to target temperatures of 34 °C, 34.5°C, and 35°C over a 3 hr period. After arriving at their target temperature, the temperature was held constant for 3 hr and then cooled back to 30 °C over an additional hour. These experimental ramps were repeated for a total of two ramps across 2 days. On the third day, tissue was airbrushed into RNALater until transport back to Hopkins Marine Station (Pacific Grove, CA).

Symbiont density was quantified using a Guava EasyCyte flow cytometer. Briefly, coral tissue was centrifuged and pelleted, the RNALater supernatant was removed, and the pellet was resuspended in 0.1 % SDS. Tissue was then homogenized using a rotostat (Thermo Fisher Scientific) and needle-sheared (10 aspirations through a 25 G needle). Samples were then diluted 1:200 and measured in triplicate on the flow cytometer (see supplemental appendix for gating protocols). Symbionts were identified based on chlorophyll fluorescence, and events with a sufficiently large forward scatter were classified as symbiont-free coral cells. The symbiont load for each replicate was calculated as symbiont-containing cells divided by the total cell count. Symbiont load was subsequently used for statistical analyses to identify bleaching-resistant colonies and to examine relationships between bleaching resistance, symbiont load, growth rate, and temperature profile of each reef. See the supplemental appendix for a more detailed description of the Materials and methods used in this study.

## Acknowledgements

The authors would like to acknowledge the staff and boat operators of the PICRC, as well as Julien Ueda, Mica Chapuis, Bowen Jiang, Callan Hoskins, Colin Hyatt, and Mehr Kumar for their assistance in

the field. Furthermore, the authors would like to thank the Aimeliik, Kayangel, Koror, and Ngarchelong state governments for their support of this project.

## Additional information

### Funding

| Funder | Grant reference number | Author |
|---|---|---|
| National Science Foundation | OCE-1736736 | Stephen R Palumbi |
| Stanford University Office of Development | | Stephen R Palumbi |

The funders had no role in study design, data collection and interpretation, or the decision to submit the work for publication.

### Author contributions

Brendan Cornwell, Conceptualization, Data curation, Formal analysis, Investigation, Methodology, Validation, Visualization, Writing - original draft, Writing - review and editing; Katrina Armstrong, Nia S Walker, Investigation, Methodology, Writing - original draft, Writing - review and editing; Marilla Lippert, Victor Nestor, Investigation; Yimnang Golbuu, Conceptualization, Funding acquisition, Resources; Stephen R Palumbi, Conceptualization, Data curation, Formal analysis, Funding acquisition, Investigation, Methodology, Project administration, Resources, Supervision, Validation, Visualization, Writing - original draft, Writing - review and editing

### Author ORCIDs

Brendan Cornwell (ID) http://orcid.org/0000-0001-7839-8379
Nia S Walker (ID) http://orcid.org/0000-0002-6314-0436

### Decision letter and Author response

Decision letter https://doi.org/10.7554/eLife.64790.sa1
Author response https://doi.org/10.7554/eLife.64790.sa2

## Additional files

### Supplementary files

• Supplementary file 1. Physical properties of reefs and average heat retention statistics.

• Supplementary file 2. Temperature extremes by geographic region. Average percent time spent per reef above 31–35°C (out of 35,764 observations) in the five geographic regions in this study. Values per individual reef can be found in *Supplementary file 1*.

• Transparent reporting form

• Source data 1. Complete_Colony_By_Colony.data.csv.

### Data availability

Temperature data have been deposited in the BCO-DMO database (https://www.bco-dmo.org/dataset/772445), all other data generated or analysed during this study are included in the manuscript and supporting files.

The following dataset was generated:

| Author(s) | Year | Dataset title | Dataset URL | Database and Identifier |
|---|---|---|---|---|
| Palumbi S | 2019 | Water temperature records for Acropora hyacinthus coral colonies located in either patch or fore reefs of the Palau Archipeglo from November 2017 to July 2018 | https://doi.org/10.1575/1912/bco-dmo.772445.1 | Biological and Chemical Oceanography Data Management Office, 10.1575/1912/bco-dmo.772445.1 |

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

# Appendix 1

## Large-scale mapping of heat tolerance

Our goal was to characterize heat tolerance in corals across many reefs and habitats by sampling and testing a large number of corals. Such maps are highly valuable for understanding the range of environmental conditions that a local coral population can respond to, and they can lead to key mechanistic studies of the underlying machinery of stress tolerance. However, their very nature presents logistical challenges and tradeoffs in study design that may be general to future efforts. In Palau, 100 s of reefs are accessible from the southern location of the PICRC. However, northern reefs and south-western reefs require 60–90 min transit time as compared to 15 min for eastern reefs. As a result, corals from different locations experienced different levels of travel stress, even after we minimized stress by careful, standardized transport and recovery conditions. This condition placed particular value on monitoring the health of controls. An additional challenge is the tradeoff between the number of corals or reefs sampled and the number of replicates per coral tested. This is especially an issue because we held the recovery time and time in lab conditions constant (all corals were fully tested within 3 days of collection). In our study we reduced replication within colonies (five fragments tested) and increased replication across colonies (400) and reefs (39). As a result, our study is strongest in determining geographic and regional patterns.

## Experimental protocol

### Site selection and temperature measurements

We selected 10 coral colonies per reef from 39 reefs distributed around the Palauan archipelago in 2017. In order to fully capture the range of potential phenotypic responses to warming, we chose reefs that encompass a wide range of naturally occurring water temperatures around the island. Reefs also ranged in size from approximately 5000 $m^2$ to >100,000 $m^2$, and in depth ranging from ca. 1–8 m at low tide. We selected 15 reefs in the northern lagoon of Babeldaob Island, where surface waters are regularly replaced by cooler oceanic waters flushing in with the tides (*Skirving et al., 2005*), and 15 reefs in the southern lagoon where water is retained for longer periods and has the potential to warm to higher temperatures. We selected patch reefs in grouped clusters that typically contained at least three reefs that were near one another. Additionally, we selected five fore reefs near each of the northern and southern patch reef regions (although we could not re-sample one of the northern fore reefs to run the bleaching assay), for a total of 39 reefs.

To identify 10 colonies on each reef, we haphazardly swam transects from the exterior to the interior of each patch reef to capture the full range of abiotic and biotic conditions at each location. We selected fore reef colonies by swimming parallel to the reef crest, just inside the edge of each reef. We tagged each colony with a unique numeric identifier (PA1-400) and recorded its location using a handheld GPS (Garmin, Lenexa, KS). Additionally, to characterize the thermal environment, we deployed a HOBO temperature logger (OnSet Computing, Bourne, MA) recording at 10 min intervals to all odd-numbered colonies (five per reef). Temperature data were collected at all locations from November 8, 2017, to July 20, 2018 (see *Supplementary file 1* for temperature records).

## Fragment collection and stress tank protocol

We experimentally stressed fragments from each of the 400 corals that we tagged and monitored for this study. We clipped medium sized fragments (approximately 8 cm width) from the edge of each colony using garden clippers, loosely packaged the fragments in bubble wrap and stored them in a cooler to be transported to the PICRC. Upon return, fragments were placed in a flowing seawater system at ambient temperatures where they recovered from transport overnight. The following day, we clipped the larger fragments into five smaller fragments and placed them in our experimental warming system. All fragments were approximately the same size and were treated in the same way, controlling for wounding effects.

Each of the 10 L experimental tanks was independently controlled using a 300 W heater and two chillers (Nova Tec, Baltimore, MD), and had fresh seawater inflow at all times (ca. one volume change in 2–3 hr) in addition to an aquarium pump ( ~280 L $hr^{-1}$) to increase flow around the fragments. Tanks were illuminated on a 12 hr light:dark cycle using LED light fixtures (ca. 22–66 μmol $m^{-2}$ $s^{-1}$). Three experimental treatments ramped a fragment from each colony to temperatures of 34 °C, 34.5 °C, or 35 °C, with two additional control tanks that did not ramp but were maintained at 30 °C for the duration of the experiment. Each experiment consisted of two ramping

cycles over the course of 2 days. Ramps began at 10:00 AM and increased temperature for 3 hr until they reached their target temperature at 1:00 PM. Heating lasted for 3 hr (until 4:00 PM) at which point chillers cooled each tank back to 30 °C, typically within 30 min. When tanks were not ramping, they maintained a constant temperature of 30 °C. After the second heat cycle, corals were held at 30 °C overnight before preservation. We preserved coral tissue by removing tissue from each fragment using an airbrush loaded with seawater. We then centrifuged the tissue for 5 min at 5000 $g$, removed the supernatant and resuspended the slurry in 2 mL of RNALater. We stored these samples at 4 °C for approximately 24 hr, after which time we transferred them to −20 °C until shipping to Hopkins Marine Station.

We visually evaluated coral fragments at 8:00 AM each day of the experiment (one observation before each ramp and a final observation the day following the second ramp for a total of three observations). We visually assessed coral fragments using a five-point scale that consisted of the following categories: 1 – no bleaching, 2 – slight bleaching visible, 3 – moderately discolored, clearly bleaching, 4 – severe bleaching, with some color remaining, 5 – no color, completely bleached. We evaluated each fragment using two scorers who were required to agree on a score for each fragment during each assessment. We also took photographs of each tank immediately following the visual bleaching score (VBS) assessment for confirmation in cases where the visual score disagreed with symbiont quantification by flow cytometry (described below).

## Quantification of symbiont load

In order to more precisely quantify the symbiont density in each fragment after experimental heating, we used a Guava EasyCyte HT (Millipore, Burlington, MA) flow cytometer to count symbiont cells. To prepare each sample, we centrifuged 500 µL of each cell suspension sample plus 500 µL filtered seawater at 15,000 $g$ for 5 min, removed the supernatant, and resuspended each pellet in 300 µL 0.01 % SDS solution. We homogenized each sample using a PowerGen rotostat (Thermo Fisher Scientific, Waltham, MA) set at the highest setting for 5 s. We then needle-sheared each sample through a 25 G needle 10 times to ensure cells were not clumped together. Each sample was then diluted by 1:200 in 0.01 % SDS and run in triplicate on the flow cytometer. We gated sample counts on the forward scatter channel, counting all events exceeding $10^2$ fluorescent units. Additionally, we counted all events that exceeded $10^4$ fluorescence units on the 690 nm detector (which can detect the autofluorescence of chlorophyll) as a symbiont count. After subtracting events we detected in the negative control (0.01 % SDS), we calculated the proportion of symbionts as the total number of symbiont events divided by the total number of events above the forward scatter gate. For a subset of samples, we also quantified the total protein content of the homogenized (undiluted) sample using a Pierce BCA protein assay (Thermo Fisher Scientific, Waltham, MA) as per protocols in *Krediet et al., 2015*. We measured the absorbance of these samples in triplicate on a Tecan Plate Reader at 562 nm. We note that while cell count values may not represent absolute numbers of cells within tested colonies (e.g., due to cell fragmentation during tissue airbrushing and preservation, or Guava cell counting error), values do confidently represent relative differences between samples.

## Analysis

### Geographic temperature variation

Previous work found that oceanic waters routinely flush through the interior of northern fore reefs during tidal cycles (*Skirving et al., 2005*), which might make these environments colder than southern interior waters. Similarly, we also expected fore reef environments that are exposed to cooler upwelling conditions to have fewer extreme temperature events than protected patch reef environments. The thermal profiles that we recovered from loggers deployed on the same reef were highly correlated; we therefore used the average thermal profile for each reef for these analyses. After trimming the temperature measurements to begin and end at the same time points, we tested these predictions by comparing the mean number of time events above 31 °C in patch reefs compared to fore reefs using a one-tailed t-test. Other temperature thresholds resulted in similar rankings. We then conducted a comparison between northern and southern patch reefs using a one-tailed t-test with the prediction that northern reefs should experience fewer high temperature events than southern reefs.

## Geographic patterns of bleaching resistance

We examined whether the rate of bleaching resistance over the course of the 2-day experiment was dependent on geographic region (northern versus southern reefs). We normalized the reduction in symbiont density for each temperature treatment by dividing the density of symbionts remaining after heat stress by the density of two averaged control treatments. We then used these standardized values of symbiont retention to assess the degree to which corals bleached during heat stress.

We also estimated the temperature at which each colony lost 50 % of its original symbionts. To do this we assumed a linear decline in symbiont density from the initial symbiont proportions in the control treatment to the final proportions in the three temperature treatments. From this linear model we then estimated the temperature treatment that would be necessary to careferencse the loss of half of the initial symbiont population.

We tested if bleaching-resistant colonies were evenly distributed across reefs in Palau. One prediction, if the thermal environment selects for heat-resistant colonies or results in acclimation to increased temperature, is that bleaching-resistant colonies should reside in warmer environments. We therefore assessed whether bleaching responses were correlated with the number of extreme temperature events on each reef. Because data loggers deployed to each reef were highly correlated (see above), we used the average number of time points above 32 °C for all loggers on a reef to represent their exposure to thermally challenging conditions. We also conducted these analyses using increasing thresholds of 33°C and 34°C, but exposure to these temperatures was too rare on many reefs (see *Figure 2—figure supplement 4* for the relationship between the number of extreme thermal events and the number of bleaching-resistant colonies for each threshold value).

To test the relation of local heating to heat resistance, we used a fully factorial linear model where the average retention was the dependent variable, and the number of extreme temperature events on a reef, the area of the reef (excluding fore reef sites), and the average depth of the colonies on each reef were the independent variables.

## Supplementary results

### Cell count and visual score data

We tested 361 colonies across 39 reefs. We removed 59 colonies from the data set for which we had only one control value (n = 30) or had one or less heat treatment values (n = 30). Of the original 361 colonies, the controls of 109 individuals showed moderate bleaching or worse after 2 days in the test tanks. These colonies may have been particularly stressed by transport and were removed from the data set.

For 33 of the original colonies, retention scores were anomalously higher than 1.0, meaning the heated branches showed more symbionts than the controls. Because we used averaged pairs of control in our retention scores, we modeled the variance expected for such pairs at a range of mean symbiont loads, using observed variances between controls. We used this variance to estimate the 95 % confidence limits for the retention scores for unheated colonies: this suggested that retention values above 1.8 are unexpected by intra-colony branch variance alone. As a result, we removed 15 colonies for which the heated branches showed anomalously higher density of symbionts than did the controls (ratio range 4.4–1.8) and retained 31 colonies with retention scores 1.0–1.8. Most of these colonies (n = 24) showed lower than average symbiont proportions in the controls (averaging 0.06). These changes left us with a filtered cell count data set of 221 colonies (*Supplementary file 1*).

We also removed outlying measurements where the symbiont proportion substantially disagreed with the VBS for that same branch. Outliers were identified as those more than 2.5 standard deviations above the mean for that VBS (see below for measured values; expected for 0.5 % of observations). For VBS of 2, 3, 4, and 5, those limits were calculated from the data in *Supplementary file 1* to be >0.218, >0.146, >0.15, >0.10, respectively. We excluded any measurements where the VBS and cell count disagreed based on these thresholds, which excluded at least one heat treatment from 27 colonies.

Initial symbiont proportions measured by flow cytometry were highly heterogeneous in corals across the archipelago. Mean symbiont proportions in the control treatments for the 221 colonies ranged from 1.98% to 26.12% at the end of the experiment (*Figure 2A*). Symbiont

proportions generally do not differ significantly among reefs, with the exception of reefs 25 and 30 (p = 0.02 and 0.04, respectively) where they are significantly lower and reef 27 where they are significantly higher (p = 0.02; linear model predicting symbiont load with reef as a factor overall $r^2$ = 0.1197, p = 0.0053), do not differ statistically between northern (0.0979) and southern (0.0975) lagoons (t = –0.058443, df = 215.25, p = 0.9534), and do not correlate with the number of extreme temperature events (see above) that occurred on the reef for the ca. 8 months preceding experimental trials of bleaching resistance ($r^2$ = 0.008123, p = 0.1232).

We measured VBS for 1608 branches from these colonies after 1 and 2 days of heating at 30°C, 34°C, 34.5°C, and 35°C. A comparison between VBS and the proportion of symbionts measured with flow cytometry shows high correlation (*Figure 1—figure supplement 1*). Coral fragments scored to have no bleaching (VBS = 1) on average show 12 % of their cells to contain symbionts whereas totally bleached corals (VBS = 5) show 3 %. However, there is high variance in symbiont proportions within each VBS category, reflecting both the variation among non-heated colonies (*Figure 1*) and the wide variation in bleaching states that fall within each visual bleaching category.

The different rates of bleaching we observed in this study were largely due to individual reactions to heat stress on the first day of the experiment followed by much smaller reductions on day 2. First, baseline measurements across all experiments on day 0 show that on average, northern corals started the experiment with equivalent symbiont densities (mean VBS = 1.71) to their southern counterparts (mean VBS 1.63, t = –1.0856, df = 217.94, p = 0.2788). From those initial densities, the decline in VBS in northern locations between day 0 and day 1 was approximately 44.8%, while the scores of individuals from southern locations declined approximately 40.6 % by the end of day 1 (t = –1.728, df = 215.23, p = 0.08543). Northern and southern populations continued to bleach between day 1 and day 2 at nearly identical rates. Northern individuals lost an additional 7.0 % of their initial symbiont populations as measured by the VBS, while southern individuals lost 6.6 %. Cumulatively, the final symbiont population densities were higher in the North than in the South, 51.8% and 47.1%, respectively (t = –2.1424, df = 217.47, p = 0.03328).

