## [Decision Letter]

**Acceptance summary:**

This paper will be of interest to the reef restoration community. The authors use a customised rapid heat stress assay to rank corals with respect to their bleaching level and back up visual observations with laboratory assays. This permitted an important spatial analysis of heat tolerance, resulting in the insight that bleaching-tolerant corals can be found everywhere, although they are more abundantly on warmer reefs.

**Decision letter after peer review:**

[Editors’ note: the authors submitted for reconsideration following the decision after peer review. What follows is the decision letter after the first round of review.]

Thank you for submitting your work entitled "Widespread variation in heat tolerance of the coral *Acropora hyacinthus* spanning variable thermal regimes across Palau" for consideration by *eLife*. Your article has been reviewed by 3 peer reviewers, one of whom is a member of our Board of Reviewing Editors, and the evaluation has been overseen by a Senior Editor. The following individual involved in review of your submission have agreed to reveal their identity: Natalia Andrade (Reviewer #3).

Our decision has been reached after consultation between the reviewers. Based on these discussions and the individual reviews below, we regret to inform you that your work will not be considered further for publication in *eLife*.

Reviewers agree on the interest your paper may raise, especially with the global warming research community. The dataset you assembled is superb, even more so since, as we appreciate, fieldwork and experiments in the tropics is not easy. However, the reviewers agreed that the analyses and conclusions are weak; in part by not showing any genetic (heritable) results, despite your acknowledgement that you have obtained RNA. The reviewers offer what we hope is valuable input in the comments below.

*Reviewer #1:*

Using a series of direct field measures and an experiment, Cornwell et al. linked variability in coral's heat tolerance to environmental factors across Palau. They found that phenotypic plasticity was larger within sites than between them. More important, heat-tolerant corals are widespread, despite being more common in warmer parts of the island. This work was framed in an important (and impressive) premise, namely, conservation strategies for future ecosystems should target populations that are resilient to climate change stressors.

*Reviewer #2:*

I was excited to read this manuscript which I expected to be focussed on describing the patterns and drivers of variation in heat/bleaching tolerance among corals sampled across la large number of reefs in Palau. I was disappointed to find that the major focus at least in the first half of the results was on the metrics of scoring heat/ bleaching tolerance. The introduction set the scene beautifully but the tread was somewhat lost in the M and M where spatial variation in temperature regimes, justifications for thresholds were omitted and instead a strong focus on the various measures of heat/bleaching tolerance was presented. Further, this analysis was described in a quite confusing way – stating the approach in the M and M and then going through its justification in the Results section – and with what appears to be contradictions in thresholds, etc. I found the results starting Figure 3 to be a lot more interesting but by the time I got to this stage I was confused about the measures used, dubious whether they could actually be measured accurately, whether they were correlated and so on, and unclear about the temperature regimes among reefs, the justification for temperature thresholds and so on. There is a lot of excellent stuff here – but it gets lost in less interesting analyses. I suggest a very critical edit is needed to describe what was done and why this was justified to form the basis for the focus of the paper that should be the geographical analysis of heat/bleaching tolerance.

*Reviewer #3:*

I think this study use original methods to demonstrate the immense variation of coral response to short heat stress. The integration of the geographical component allowed to show the existence of resisting corals across Palau reefs, which might partially guide any attempted of appropriate reef restoration programs. It reads well and I think with some changes the paper will provide valuable information to other researchers and will help discover new knowledge gaps.

I have some general comments:

1. An explicit definition of what a heat resistant coral or a bleaching resistant coral is in this study is needed from the beginning of the manuscript. There is a range of studies claiming to work on heat resistant or bleaching resistant corals. Some define this resistance as surviving the bleaching (even if they have bleached) and in other cases, like this study, low bleaching threshold. I think stating clearly what you are considering a resisting coral will avoid misunderstandings. Setting a definition would help the reader to put more perspective into the discussion.

2. I think you should mention how you took into account the size of the coral fragment when analyzing the proportion of zooxs or protein analysis. Most of the analysis in the study is done using that information so I consider that it is important to clarify that you measure the surface of the fragments or that you try to correct for size. I cannot see any information related to that in the document.

3. It is necessary to acknowledge at some point in your manuscript that the fact that the corals did not bleach or show a low score of bleaching after the 2 days treatment does not prove that those colonies will resist bleaching in a natural environment or longer stress. Two days of treatment help to distinguish the different responses to acute stress and revealed phenotypic diversity. Unfortunately, this experiment cannot give many insides on how these corals will respond after weeks of it. Which is ok, but it might be better to be more straight forward about that.

[Editors’ note: further revisions were suggested prior to acceptance, as described below.]

Thank you for submitting your article "Heat tolerance and symbiont load are associated with growth tradeoffs in the coral *Acropora hyacinthus* across Palau" for consideration by *eLife*. Your article has been reviewed by 2 peer reviewers, and the evaluation has been overseen by Meredith Schuman as the Reviewing and Senior Editor. The following individuals involved in review of your submission have agreed to reveal their identity: Natalia Andrade (Reviewer #1); Line Bay (Reviewer #2).

The reviewers have discussed their reviews with one another and the Editor, and the Editor has drafted this to help you prepare a revised submission.

Essential Revisions:

Here, we list the essential revisions to which the authors should respond in a revision and point-by-point response letter. The original reviews are then appended in full for the authors' information, but the original reviews do not require a point-by-point response.

1) Temper claims. The approach proposed by the authors is a simple and powerful way to determine potential heat resistance of colonies which is supported by a geographically large dataset which reveals that heat tolerant coral colonies are widespread, and further identifies a previously unknown relationship between growth rate and symbiont load. However, the method is not sufficient to identify colonies for restoration; rather, it is a sort of first screening step to determine which colonies should be more closely investigated. See also Essential Revision 2, which will support the tempering of claims and focusing impact of the current study.

2) Revise the main text to accommodate publication as a short report and increase the likely impact on the field. We believe this can be done quickly and efficiently according to the recommendations below; first, see the reasoning behind this request.

Reasoning: It is the function of short reports to set the stage on an important topic using a compelling and well-conducted study, but of limited scope in terms either of geography, time, or detail. Here, the scope is limited in its biological detail, but the authors convincingly argue that the establishment of the approach described here already entails sufficient information and complexity for a stand-alone article and is important to set the stage for future work. The reviewers agree that the main novelty and importance of this paper is to demonstrate the distribution of heat-resistant colonies and propose approaches for screening, which must be combined with more detailed analsyses, some of which the authors already have in progress for future publications, in order for the potential of the approach to be realized for coral restoration. This is perfect for a short report, and reformatting the paper as a short report would eliminate the expectation that the authors' genetic and symbiont characterization datasets be included here.

Recommendations:

– There are only four figures and it seems to me that Figures 1 and 2 could be combined; I think that the authors could easily reduce to max 4 main display items by doing this, or by assigning the tables as figure supplements rather than main tables.

– The main challenge will be to reduce the text length; the authors do not need to keep strictly to the recommended length of 1500 words, but they could substantially reduce length by shortening the introduction and integrating results and discussion, which would also eliminate some of the claims which reviewers agree are not well supported by this study alone. The conclusions should focus on (1) a clear statement of the importance of this study as well as (2) a recommendation for how to build on this study to achieve coral conservation and restoration. Please note comments from Reviewer 1 regarding for example the substantial stress of healing which must be considered, as well as considerations of coral and symbiont diversity which the authors already plan to address in an upcoming study.

– The authors may also consider moving detailed but essential material and methods information to an appendix formatted as a how-to guide for using their approach to screen corals, which might serve as a detailed field protocol for other researchers to adopt, and increase the impact of the authors' findings and approach.

*Reviewer #1:*

In this study, the authors use a simple but powerful approach, heat stress and symbiont counts, to determine the potential heat resistance colonies in Palau reefs. It is the experimental design with a large geographic scale that covered 39 reefs with many unique temperature patterns, habitat variables and the high number of replicates that allows the authors to reveal a connection between heat resistance colonies and symbiont load.

This method can certainly be used as a first step into determining heat-resisting colonies. However, as the authors acknowledge, many more traits need to be taken into consideration before choosing a colony for restoration. These considerations cannot be met with the methods proposed by the authors, therefore this technic although than informative, needs to be coupled with other methods.

A key finding in this study is also the relationship between symbiont load and growth rate. As they clearly state, this is a crucial aspect that needs to be taken into account when choosing which colonies to use for reef restoration purposes.

I still think it is a nice study, with very interesting results. However, the results require more testing or integration of more data in order to through more concrete answers. It would be much more powerful having the information about the symbionts types and/or colony genetic variation data. Without that extra evidence, as far as I know, the paper is out of the scope of *eLife*. However, I think it can be publishable in another journal.

One recommendation for future studies is to increase the period of acclimatisation/wound healing. The energetic effort of healing is a. strong extra stressor and, even if you had controls treated the same way, it might have compromised the resistance of some of your fragments.

*Reviewer #2:*

This is a lovely manuscript and I appreciate the edits you have made based on the previous review process that I participated in. The text is clear and easy to follow. Your arguments are well laid out and your results are super important for the field.

---

## [Author Response]

[Editors’ note: the authors resubmitted a revised version of the paper for consideration. What follows is the authors’ response to the first round of review.]

Reviewers agree on the interest your paper may raise, especially with the global warming research community. The dataset you assembled is superb, even more so since, as we appreciate, fieldwork and experiments in the tropics is not easy. However, the reviewers agreed that the analyses and conclusions are weak; in part by not showing any genetic (heritable) results, despite your acknowledgement that you have obtained RNA. The reviewers offer what we hope is valuable input in the comments below.Reviewer #2:I was excited to read this manuscript which I expected to be focussed on describing the patterns and drivers of variation in heat/bleaching tolerance among corals sampled across la large number of reefs in Palau. I was disappointed to find that the major focus at least in the first half of the results was on the metrics of scoring heat/ bleaching tolerance. The introduction set the scene beautifully but the tread was somewhat lost in the M and M where spatial variation in temperature regimes, justifications for thresholds were omitted and instead a strong focus on the various measures of heat/bleaching tolerance was presented. Further, this analysis was described in a quite confusing way – stating the approach in the M and M and then going through its justification in the Results section – and with what appears to be contradictions in thresholds, etc. I found the results starting Figure 3 to be a lot more interesting but by the time I got to this stage I was confused about the measures used, dubious whether they could actually be measured accurately, whether they were correlated and so on, and unclear about the temperature regimes among reefs, the justification for temperature thresholds and so on. There is a lot of excellent stuff here – but it gets lost in less interesting analyses. I suggest a very critical edit is needed to describe what was done and why this was justified to form the basis for the focus of the paper that should be the geographical analysis of heat/bleaching tolerance.

We agree that the dual nature of the results based on Visual Bleaching Score and symbiont cell fraction obscured the results we were striving to convey, and so we largely dropped the Visual Bleaching Score data from the paper. However, we continue to use Visual Bleaching Score as a real-time assay of coral health during our experiments, and exclude experiments in which the controls began to change in bleaching state. This one use of Visual Bleaching Scores is explained simply in the results as a way of shoring up the quality of the data set. Other aspects of the relationship between Visual Bleaching Scores and symbiont fraction are being presented in a different methodological paper.

Reviewer #3:I think this study use original methods to demonstrate the immense variation of coral response to short heat stress. The integration of the geographical component allowed to show the existence of resisting corals across Palau reefs, which might partially guide any attempted of appropriate reef restoration programs. It reads well and I think with some changes the paper will provide valuable information to other researchers and will help discover new knowledge gaps.I have some general comments:1. An explicit definition of what a heat resistant coral or a bleaching resistant coral is in this study is needed from the beginning of the manuscript. There is a range of studies claiming to work on heat resistant or bleaching resistant corals. Some define this resistance as surviving the bleaching (even if they have bleached) and in other cases, like this study, low bleaching threshold. I think stating clearly what you are considering a resisting coral will avoid misunderstandings. Setting a definition would help the reader to put more perspective into the discussion.

We agree that this is an important issue. Heat resistance is a continuous variable and is relative to local, regional, or species-wide means, depending on the context. We added this to the beginning of the section on geographic patterns: “We focus on corals that are in the upper 25% of the heat resistance rankings and test for patterns in distribution and abundance of this set. These most heat resistant colonies were found at many locations…”

2. I think you should mention how you took into account the size of the coral fragment when analyzing the proportion of zooxs or protein analysis. Most of the analysis in the study is done using that information so I consider that it is important to clarify that you measure the surface of the fragments or that you try to correct for size. I cannot see any information related to that in the document.

The standardization that we used in this study was to measure symbiont cells as a function of all cells in a suspension. We took into account the size of the fragment by airbrushing tissue from the entire piece, and assaying the fraction of cells in that sample that were symbionts. We updated the Materials and methods to reflect this: “To measure symbiont cell density, we removed tissue from the entirety of each fragment using an airbrush loaded with seawater. We then centrifuged the resulting suspension for five minutes at 5,000 g, removed the supernatant and resuspended the slurry in 2

ml of RNALater. We stored these samples at 4°C for approximately 24 hours, after which time we transferred them to -20°C until shipping to Hopkins Marine Station.”

3. It is necessary to acknowledge at some point in your manuscript that the fact that the corals did not bleach or show a low score of bleaching after the 2 days treatment does not prove that those colonies will resist bleaching in a natural environment or longer stress. Two days of treatment help to distinguish the different responses to acute stress and revealed phenotypic diversity. Unfortunately, this experiment cannot give many insides on how these corals will respond after weeks of it. Which is ok, but it might be better to be more straight forward about that.

Agreed, a section about this is now added to the discussion.” One issue is that the heat-pulse experiment we use here results in very different bleaching phenotypes but these phenotypes might be different when bleaching occurs in natural conditions. Long exposure to low level temperature, for example, might spark a slightly different bleaching reaction, and few studies compare the reaction of a coral to different conditions (Grotolli et al. 2020). In one study, Morikawa and Palumbi (2019) tested four species of coral with the same heat-pulse experiment used here, and then scored the reaction of these same genotypes to a natural bleaching event. They found that colonies showing higher experimental bleaching resistance also showed higher natural bleaching resistance.”

[Editors’ note: what follows is the authors’ response to the second round of review.]

Essential Revisions:Here, we list the essential revisions to which the authors should respond in a revision and point-by-point response letter. The original reviews are then appended in full for the authors' information, but the original reviews do not require a point-by-point response.1) Temper claims. The approach proposed by the authors is a simple and powerful way to determine potential heat resistance of colonies which is supported by a geographically large dataset which reveals that heat tolerant coral colonies are widespread, and further identifies a previously unknown relationship between growth rate and symbiont load. However, the method is not sufficient to identify colonies for restoration; rather, it is a sort of first screening step to determine which colonies should be more closely investigated. See also Essential Revision 2, which will support the tempering of claims and focusing impact of the current study.

We have revised the manuscript with this recommendation in mind. Notably, we have made points to emphasize that this is the first step in the process of determining the genetic basis of this trait and propose follow-up experiments that would begin to assess the permanence of this trait.

2) Revise the main text to accommodate publication as a short report and increase the likely impact on the field. We believe this can be done quickly and efficiently according to the recommendations below; first, see the reasoning behind this request.Reasoning: It is the function of short reports to set the stage on an important topic using a compelling and well-conducted study, but of limited scope in terms either of geography, time, or detail. Here, the scope is limited in its biological detail, but the authors convincingly argue that the establishment of the approach described here already entails sufficient information and complexity for a stand-alone article and is important to set the stage for future work. The reviewers agree that the main novelty and importance of this paper is to demonstrate the distribution of heat-resistant colonies and propose approaches for screening, which must be combined with more detailed analsyses, some of which the authors already have in progress for future publications, in order for the potential of the approach to be realized for coral restoration. This is perfect for a short report, and reformatting the paper as a short report would eliminate the expectation that the authors' genetic and symbiont characterization datasets be included here.Recommendations:– There are only four figures and it seems to me that Figures 1 and 2 could be combined; I think that the authors could easily reduce to max 4 main display items by doing this, or by assigning the tables as figure supplements rather than main tables.

We have combined figures and moved others to the supplementary material and now have 3 figures and a table.

– The main challenge will be to reduce the text length; the authors do not need to keep strictly to the recommended length of 1500 words, but they could substantially reduce length by shortening the introduction and integrating results and discussion, which would also eliminate some of the claims which reviewers agree are not well supported by this study alone. The conclusions should focus on (1) a clear statement of the importance of this study as well as (2) a recommendation for how to build on this study to achieve coral conservation and restoration. Please note comments from Reviewer 1 regarding for example the substantial stress of healing which must be considered, as well as considerations of coral and symbiont diversity which the authors already plan to address in an upcoming study.

We have substantially reduced the number of words in the document to ~2500 in the main text (excluding references and figure legends). Furthermore, we have tempered claims throughout the text, including in the conclusions.

– The authors may also consider moving detailed but essential material and methods information to an appendix formatted as a how-to guide for using their approach to screen corals, which might serve as a detailed field protocol for other researchers to adopt, and increase the impact of the authors' findings and approach.

We have moved the majority of our methods section as well as additional analyses into the supplemental. In doing so we have added more details to the methods so that this method can be easily replicated, as well as described our reasoning and constraints in conducting the experiment (Supp. l. 3-18).